# Peer review of "Irradiation Damage Independent Deuterium Retention in WMoTaNbV"

_materials, 2022, doi:10.3390/ma15207296_

Round 1

Reviewer 1 Report

What is the expected amount of irradiation damage to the actual fusion reactor wall? (displacements per atom)

The following text is written on lines 271-272.

"irradiation does not seem to produce defects in WMoTaNbV that trap hydrogen" (*)

In this experiment, is the degree of the ion irradiation damage in the WMoTaNbV alloy the same as that in the actual fusion reactor wall?

(If the degree in this experiment is much less than that to the actual fusion reactor wall,)

If the degree of irradiation damage in the WMoTaNbV alloy is increased to the same as that to the actual fusion reactor wall, does the above statement (*) hold true?

I consider this answer to be an important matter because it is highly relevant to the purpose of this study.

For example, if this study has proved the following matter "the trapping energy in WMoTaNbV irradiated to the same degree as the actual fusion reactor wall is the same as the trapping energy in unirradiated WMoTaNbV",

I believe that this study is meaningful.

Author Response

1) In this experiment, is the degree of the ion irradiation damage in the WMoTaNbV alloy the same as that in the actual fusion reactor wall?
Answer: The damage produced by this experiment is much lower than that of an actual reactor wall. The research on this topic is at its early stage and this study is focusing on the fundamental differences in D retention and trapping mechanisms in two materials.

2) If the degree of irradiation damage in the WMoTaNbV alloy is increased to the same as that to the actual fusion reactor wall, does the above statement (*) hold true?

Answer: This question is undoubtedly important, but its scope is too wide to be answered certainly based on presented results. The irradiation energy is relevant to fusion, but the experimental setup is much simplified
compared to the actual environment of the reactor. In the reactor there are many isotope species present with a range of energy along with a variety of other species from the fuel and reactor wall. The damage created in
such environment is more severe and it may interact with hydrogen differently. To answer this question properly, more work is needed both on 1) a fundamental level investigating the basics of phenomena, as we do in this
study, and also 2) on more application focused level. However, this work does show that the trapping mechanisms are very different in WMoTaNbV alloy compared to W and it provides a solid basis for further exploration.

Reviewer 2 Report

While the purpose and significance of the study are well understood, the interpretation of the data being obtained needs to be reviewed, and the overall explanation is considered insufficient. In addition, since citations are limited, more research should be done to examine multiple references to fully examine the data, including the variability of past data.

Below is a list of areas that need to be revised.

Abstract

-->at%

Line 59:  % -->at% ? wt%?

Line 68:  (2h)-->for 2 hrs, in vacuum --> in a vacuum

Line 83 :  2h -->2 hrs (please take a space between value and unit)

Line 259: Is the word right for “solution energy”?

Line 299:  HEA  High entropy Alloy -->High Entropy Alloy

Section 3.1.

When performing the bursting calculation with respect to Figure 1, please indicate the value of the atomic threshold energy employed in the calculation and the literature on which the value is based.

Section 3.2.

Since the positron lifetime after irradiation is clearly longer in W-HEA than in W, it is not considered to be the same defect and should be Review.

Figure 2: The temperatures of D emission in W and W-HEA are completely different and the amount of emission is also different. and the amount of D emitted is also different.

Figure 4: Shouldn't you review the two main temperature regions where hydrogen is emitted, since the W-HEA has a higher temperature in the higher temperature region, and the difference in trapping energy is not considered to be 0.1 eV? Also, you should state why the overall emission is higher for W-HEA, etc.

The introduction or discussion should mention the hydrogen trapping behavior of other HEAs.

It should cite multiple data from other researchers as much as possible.

It should be checked and referred for papers of (examples:) K. Sato et al./ Nuclear Materials and Energy 9 (2016) 554-55 ; K. Sato et al. J. Nucl. Mater. 560, March 2022, 153483.

Based on the above, the summary and conclusion sections should be reviewed.

Reviewer 3 Report

The submitted manuscript investigated the deuterium retention in WMoTaNbV HEA to test the possibility of replacing the currently used W as a fusion reactor wall. It was found that the irradiation damage related H/D retention in WMoTaNbV is not significant as that in W. Some questions and comments are as follows:

 1) In the section 3.2, it was mentioned that vacancy-type irradiation damage in WMoTaNbV is at least as important as in W. However, the main conclusion is that irradiation does not produce hydrogen trapping defects in WMoTaNbV, i.e., trapping energy of irradiation induced surface defects is similar to that of bulk trapping site. However, this trapping energy is even higher than that of W (as shown in Fig. 2). Therefore, it is not convincing that WMoTaNbV could perform better than W.

 2) The discussion in line #190-192 may contradict to the explanation on Fig. 1. In Fig. 1, D diffuses fast toward the bulk interior, which is why the surface concentration is much lower than the SRIM profiles. However in line #190-192, it is written that fast D diffusion should rapidly transfer D from the implanted area to surface and eventually to vacuum without considering the possibility of fast diffusion toward bulk (i.e. diffusion to the opposite direction). It is somewhat confusing.

 3) In Fig. 4, the hydrogen release signal of HEA also displays a shoulder between 350-600°C. Can it be related to surface defects?

Author Response

1) High retention of hydrogen is not ideal considering application in fusion and based on this work it doesn’t look like WMoTaNbV would perform better than W in terms of hydrogen retention. We state this clearly in the manuscript on the lines [274-275]: “From the perspective of applicability as a fusion first wall material, the high hydrogen retention is non-ideal.” This work shows the different mechanisms being at play behind D trapping. In W the D traps are mainly monovacancies and small vacancy clusters created by irradiation, yet the similar effect is not visible in WMoTaNbV. Small clusters vacancies are also created in the surface region due to irradiation in WMoTaNbV, but D trapping occurs in the bulk. The nature of trapping sites is not investigated in this work, but it may be related to impurity atoms (such as C, N, O) introduced during the manufacturing process. Further research is needed on the causes of trapping in WMoTaNbV.

2)This is true, most of the D is diffused in the bulk of WMoTaNbV and gets trapped there. This part is meant to describe our hypothesis which, eventually, did not match the results of the experiment. This was not stated clearly in the original draft and we have made a clarification on this in lines 189-193.

3)The nature of H release signal of HEA between 350-600C is not clear, it certainly could be related to surface defects, but the fairly large signal also suggests release from shallower traps in the bulk. Dedicated studies to resolve this issue is still needed.

Round 2

Reviewer 1 Report

For the readers' convenience, I think it would be good to add to the Abstract the maximum damage density (atoms/cm3 and displacements per atom) and depth in the D-implanted WMoTaNbV alloy in this experiment.

I leave it to the readers to compare the damage density in this experiment with those of actual fusion reactor walls.

Author Response

The damage density of WMoTaNbV alloy and depth of maximum damage is now added to the Abstract.

Reviewer 2 Report

Several points should be corrected as follows

1. the important result that the trapping energy is 1.7 eV should be supplemented in the overview section to show how large it is relative to pure W.  Also, there should be a little more supplement in the introduction or in the results and discussion as to what the trap energy is for what type of defects.

2. With respect to W, what is described in lines 160 to 162 is already described in JNM, 2017. The similar values have already been shown in K. Sato's paper in JNM, 2017 (vol.496, pp.9-17). Since the W-HEA values were evaluated after obtaining almost the same values in this study, the W-HEA data should be discussed after properly citing Sato's paper. Also, doing so would make the results of this paper more credible.

3. in Figure 4, an Arrhenius plotted figure is shown, but the error of the measured data at each temperature should be stated. Also, the original data is shown, but the blue and red lines should be modified so that they precede the black line.

4. SRIM in the description of the lines in Figure 1 should be corrected to the range distribution of incident ions.

5. Information on the deuterium emission temperature is also important, so please include it in the summary (if the word limit allows) and in the summary section.

Author Response

1. The trapping energies of all components of W-HEAs are provided in the Table 2.. The obtained value of 1.7 eV is also compared to that of pure W, and W with C-interstitials, in the text:

lines 234-235 “The trapping energy to the mono-vacancies in W has a similar magnitude (1.6 eV) to the D trapping in WMoTaNbV (1.7 eV).”

lines 244-246 ”For example, the substitutional carbon has a a relatively strong trapping energy of about 1.5 eV in W [22]. This suggests that in specific environments C impurities can act as strong traps.”

2. We have addressed this concern by adding a following citations:
1) T.Troev, et al. Nucl.Instrum. 2009 vol.267 pp.535-541
2) P.E.Lhuillier, et. al. Phys Status Solidi C 2009 vol.6 pp.2329-2332.

3. Blue and red lines are now modified so that they precede the black line.
The error of temperature-fitting is of magnitude less than 0.2 °C and is negligible compared to the error that is presented in Arrhenius plotted figure.

4. The range distribution of incident ions of SRIM simulation can be found in the Figure 1. The legend is changed to indicate this more clearly. 

5. Deuterium emission temperature of W-HEA is now added to the Abstract.